# CONTROL: A CONTRASTIVE LEARNING FRAMEWORK FOR OPEN-WORLD SEMI-SUPERVISED LEARNING

## ABSTRACT

In recent years, open-world semi-supervised Learning has received tremendous attention. This is largely due to the fact that unlabeled real-world data often encompasses unseen classes – those that are not represented in labeled datasets. Such classes can adversely affect the performance of traditional semi-supervised learning methods. The open-world semi-supervised learning algorithms are designed to enable models to distinguish between both seen and unseen classes. However, existing algorithms still suffer from the problem of insufficient classification of unseen classes and may face the risk of representation collapse. In order to better address the aforementioned issues, we propose a contrastive learning framework called CONTROL that integrates three optimization objectives: nearest neighbor contrastive learning, supervised contrastive learning, and unsupervised contrastive learning. The significance of the framework is explained by theoretically proving the optimization of contrastive learning at the feature level benefits unseen classification, and the uniformity mechanism in contrastive learning further helps to prevent representation collapse. Serving as a unified and efficient framework, CONTROL is compatible with a broad range of existing open-world semi-supervised learning algorithms. Through empirical studies, we highlight the superiority of CONTROL over prevailing state-of-the-art open-world semi-supervised learning algorithms. Remarkably, our method achieves significant improvement in both unseen class classification and all class classification over previous methods on both CIFAR and ImageNet datasets.

## 1 INTRODUCTION

With the development of deep learning (Goodfellow et al., 2016), numerous significant works have been proposed in various fields such as natural language processing (Otter et al., 2020) and image recognition (Rawat & Wang, 2017). However, these advancements rely on expensive and hard-to-obtain labeled data. Therefore, the effective utilization of unlabeled data in semi-supervised learning has received increasing attention.

Traditional semi-supervised learning assumes that labeled and unlabeled data follow the same distribution. Nevertheless, maintaining this assumption is challenging in practical applications. Consequently, open-world semi-supervised learning, which contemplates more realistic scenarios, is currently gaining popularity. As shown in Figure 1, the major distinction between traditional and open-world traditional semi-supervised learning is that open-world semi-supervised learning incorporates unseen classes into consideration and emphasizes the classification of both known (seen) and unknown (unseen) classes.

In real-world situations, there are often numerous new categories that arise or have not been timely labeled by humans. As mentioned in (Cao et al., 2022), in the context of social media classification, we aim to effectively categorize both known customer groups and newly emerging user groups. Similarly, object recognition tasks in supermarkets encounter thousands of new items every day, and promptly annotating these is financially taxing (Han et al., 2020). Therefore, the significance of open-world semi-supervised learning becomes apparent.

The key to solving open world semi-supervised learning problem (Cao et al., 2022) lies in how to ensure the model achieves satisfactory performance in classifying both seen and unseen classes. Since seen classes can be learned directly and efficiently from labeled data while unseen classes can not, it

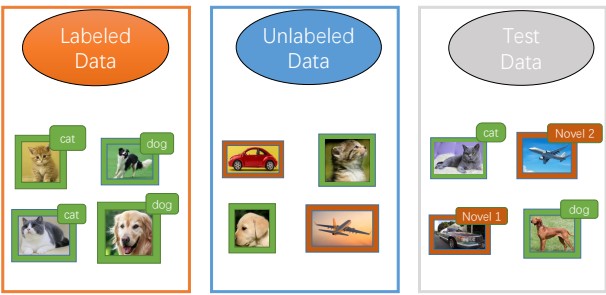

Figure 1: Open-World SSL: the unseen classes present in unlabeled data (the samples in the red box ) do not appear in the labeled data, and the unlabeled data share the same distribution with the test data. In the testing phase, both seen classes and unseen classes should be classified.

is important to prevent unseen class samples from being overfitted to seen classes. We feature out that how to select unlabeled samples to obtain efficient seen and unseen clustering results becomes the key to solving the problem. A mainstream approaches just like ORCA (Cao et al., 2022) and NACH (Guo et al., 2022) aim at making the logits vector of each unlabeled sample and its nearest neighbor as similar as possible, it is easy to see that these methods classify seen and unseen classes by selecting pairs of plausible samples and we call them BCE-based methods since BCE loss is used to optimize the logits similarity between chosen sample pairs. There are also other algorithms that consider the similarity between all unlabeled samples and use all the similarity as a guide for further optimization, and we call these other methods such as OpenLDN (Rizve et al., 2022a) and TRSSL (Rizve et al., 2022b).

In open world semi-supervised learning problem, high quality chosen samples are utilized to give the model the ability to classify both seen and unseen classes. The strategy for selecting the samples is critical, we figure out that BCE-based methods can choose sample pairs in a more flexible way. Taking NACH (Guo et al., 2022) as an example, the selection of which sample pairs need to be pulled closer can be flexibly determined through the use of filter. Conversely, in OpenLDN(Rizve et al., 2022a), the similarity function is honed by directly computing the MSE loss between feature similarity of all sample pairs and their logits counterparts. However, not all the similarity of sample pairs can efficiently guide the subsequent learning process since unseen class samples are prone to be mislabeled as seen classes in initial training phases. Therefore, BCE-based methods are more suitable for choosing high quality samples since these methods can choose samples pairs in a more flexible way.

However, despite the current flexibility demonstrated by BCE-based methods in selecting similar sample pairs, the ratio of seen-unseen pairs (the most similar sample of an unlabeled unseen classes data is from seen classes) in the chosen sample pairs still remains high as shown in Figure 2. The reason is that current BCE-based methods focus on pairwise clustering only at the logits level which could lead to poor classification performance of unseen classes. We notice that whether pairs should be aligned at the logits

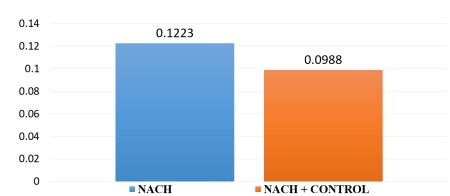

Figure 2: Ratio of seen-unseen pairs when NACH combined or not combined with CONTROL.

level depends on the similarity at the feature level. Therefore, we theoretically demonstrate that optimizing at the feature level benefits pairwise clustering at the logits level. Additionally, it is crucial to note that despite the presence of entropy loss regularization in the current open-world semi-supervised learning algorithms, there still exists a risk of representation collapse for unseen classes. Thus we introduce contrastive learning as a means to obtain superior representations for open-world semi-supervised learning and further avoid the collapse of the unseen classes. We also note that OpenCON (Sun & Li, 2022) proposed a paradigm that only utilizes contrastive learning loss to address the open-world semi-supervised learning problem. However with this learning paradigm,

existing losses widely used in semi-supervised learning (such as cross entropy loss, BCE loss) are difficult to be directly combined with OpenCON since it only focus on the optimization of the feature-level. In other word, it is incapable of sustaining continuous optimization of representations during the process of semi-supervised learning. It could be regarded as providing a superior backbone for existing open-world semi-supervised learning algorithms. Thus, a problem naturally arises:

*Can we design a unified contrastive learning framework that can further gain performance improvement specifically for open-world semi-supervised learning algorithms?*

To tackle this challenge, we propose a supervised contrastive learning loss and a nearest neighbor contrastive learning loss to enhance the classification of BCE loss. We also propose an unsupervised contrastive learning loss as a consistency regularization to further avoid unseen class collapse. Our framework is evaluated on various representative datasets. The whole framework achieves $6.4\%$ improvement in unseen classes classification and $2.1\%$ improvement in all classes classification on the CIFAR-100 dataset than the state-of-the-art method NACH (Guo et al., 2022).

To sum up, this paper makes the following three contributions:

- We proposed a simple and efficient **CONTR**astive learning framework for **O**pen World semi-supervised **L**earning **CONTROL**, which can be easily adapted to open-world semi-supervised learning algorithms.

- We theoretically demonstrate that the proposed CONTROL can not only enhance the classification of BCE loss but also avoid unseen classes collapse.

- Extensive experiments have demonstrated the effectiveness of the overall framework and its individual components.

## 2 RELATED WORK

**Traditional Semi-Supervised Learning (SSL).** Traditional SSL assumes that labeled, unlabeled, and test data are drawn from the same distribution. The mainstream of SSL algorithms can be broadly classified into entropy minimization methods (Lee, 2013; Grandvalet & Bengio, 2004), consistency regularization methods (Miyato et al., 2019; Sajjadi et al., 2016; Laine & Aila, 2017; Tarvainen & Valpola, 2017), and hybrid methods (Sohn et al., 2020; Berthelot et al., 2019; 2020; Xu et al., 2021; Zhang et al., 2021). However, traditional semi-supervised learning algorithms fail to address the open world problem with numerous unseen classes that we aim to solve.

**Open world Semi-Supervised Learning.** This paradigm is to address such a problem: in the training phase, i.e., unseen classes appear in the unlabeled data. While during the testing phase, samples from unseen classes could also appear. Open world semi-supervised learning algorithms aim at classifying both seen classes and unseen classes. Existing methods can be generally categorized into two groups: BCE-based methods, including (Cao et al., 2022) and (Guo et al., 2022); and other methods, such as (Rizve et al., 2022a) and (Rizve et al., 2022b). As mentioned earlier, optimization at the feature level is crucial for open world semi-supervised learning algorithms. However, this aspect has not received sufficient attention, which is also the biggest distinction between our proposed framework and previous algorithms.

**Contrastive Learning for Semi-Supervised Learning.** Semi-supervised learning algorithms combined with contrastive learning have also developed rapidly (Li et al., 2021; Yang et al., 2022). CACSSL (Yang et al., 2022) is a contrastive learning framework as a general confirmation bias alleviation method for pseudo-label-based SSL methods. And Comatch (Li et al., 2021) takes advantage of graph-based contrastive learning to learn better representations for corresponding classification tasks. However, none of these methods have focused on the more realistic problem of open world semi-supervised learning. Although OpenCon (Sun & Li, 2022) considered the open world semi-supervised learning problem, it is incapable of sustaining continuous optimization of representations during the process of semi-supervised learning, which is the key differentiation between OpenCon and CONTROL.

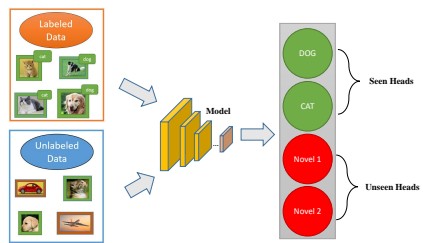

Figure 3: Model designed for open world semi-supervised learning.

## 3 PRELIMINARY AND BACKGROUND

In this section, we introduce the open-world semi-supervised learning setting and the existing BCE-based methods and representative contrastive learning loss which will be used in the next section.

### 3.1 OPEN-WORLD SEMI-SUPERVISED LEARNING

As shown in Figure 3, we are given a set of labeled data includes $n$ instances and we are also given a set of unlabeled data from an unknown distribution, which includes $m$ instances $\mathcal{D}_u = \{\mathbf{x}_1, \cdots, \mathbf{x}_m\}$. Here, $\mathbf{x} \in \mathcal{X} \in \mathbb{R}^D, \mathbf{y} \in \mathcal{Y} = \{1, \cdots, C_U\}$ where $D$ is the number of input dimension and $C_U$ is the total number of classes in unlabeled data. We use $C_L$ to represent the total number of classes in labeled data. Note that in the Traditional Semi-Supervised Learning, $C_L = C_U$ while in open-world semi-supervised learning, total number of seen classes $C_{seen} = C_L$, and total number of unseen classes $C_{unseen} = C_U \setminus C_L$.

### 3.2 REVISIT OF BCE-BASED METHODS

We now describe how BCE-based methods design optimization objectives for open-world semi-supervised learning. We use $\mathcal{L}_{\text{OWSSL}}$ to represent the overall optimization objective for existing BCE-based methods, we have

$$\mathcal{L}_{\text{OWSSL}} = \mathcal{L}_{\text{CE}} + \mathcal{L}_{\text{BCE}} + \mathcal{L}_{\text{Entropy}} + \mathcal{L}_{\text{Balance}}, \tag{1}$$

$\mathcal{L}_{\text{CE}}$ is usually defined as cross-entropy loss for labeled data, and $\mathcal{L}_{\text{Entropy}}$ aims at regularizing the predictive distribution of the training data to be close to a prior probability distribution to prevent the model from classifying all unseen classes into a single class. To enhance model's ability of unseen classes classification, $\mathcal{L}_{\text{BCE}}$ is defined as $\mathcal{L}_{\text{BCE}} = -\frac{1}{m+n} \sum_{\mathbf{x}_i \in \mathcal{D}_l \cup \mathcal{D}_u} \log \left( p\left(\mathbf{x}_i\right)^\top p\left(\widetilde{\mathbf{x}}_i\right) \right)$, where $\widetilde{\mathbf{x}}_i$ is the most similar sample to $\mathbf{x}_i$ in this batch. In NACH, a filtering mechanism is proposed to further avoid seen-unseen pairs. To balance the learning pace of seen classes and unseen classes, a variety of $\mathcal{L}_{\text{Balance}}$ are proposed, in Cao et al. (2022) $\mathcal{L}_{\text{Balance}}$ is a margin cross-entropy to mitigate the classification pace of seen classes while in Guo et al. (2022), an adaptive threshold based on pseudo-label selection is proposed to accelerate the learning pace of unseen classes.

### 3.3 CONTRASTIVE LOSSES

We introduce the generalized contrastive Loss based on the L2-normalized feature embedding $z = \phi(\mathbf{x}; \theta)$. Then the following loss function is calculated for each sample x:

$$\mathcal{L}_\phi(x; \tau, \mathcal{P}(x), \mathcal{N}(x)) = -\frac{1}{|\mathcal{P}(x)|} \sum_{z^+ \in \mathcal{P}(x)} \log \frac{\exp\left(z^\top \cdot z^+/\tau\right)}{\sum_{z^- \in \mathcal{N}(x)} \exp\left(z^\top \cdot z^-/\tau\right)}, \tag{2}$$

where $\mathcal{P}(x)$ is the positive set of embeddings w.r.t $z^+$, and $\mathcal{N}(x)$ is the negative set of embeddings w.r.t $z^-$ and $\tau > 0$ is the adjustable temperature parameter. Such a loss function pulls together the anchor $x$ and its positive set $\mathcal{P}(x)$ and pushes apart samples from its negative set $\mathcal{N}(x)$.

## 4 PROPOSED CONTROL FRAMEWORK

Despite the success of BCE-based methods, we observe that these methods still encompass two major challenges. The first is to correctly classify same-class unseen pairs. Note that there are no labeled samples available for the unseen classes. The second challenge for the BCE-based methods is to avoid the collapse of the unseen classes.

In this paper, we intend to solve these two major challenges by introducing additional alignment and uniformity on the feature level. Specifically, in Section 4.1, we show that BCE loss suffers from incorrect unseen pairs, whereas contrastive loss is able to make correct alignments. Thus, we introduce the supervised contrastive loss for the seen classes and nearest neighbor contrastive loss for all classes, which aligns both the seen and unseen pairs on the feature embedding space. Then in Section 4.2, we demonstrate that the uniformity term in contrastive losses serves as an additional term to avoid unseen class collapse. In Section 4.3, we show that unsupervised contrastive loss serves as a consistency regularization on the feature level and also guarantees consistency on the logit level. Finally, in Section 4.4, we introduce the overall method of our CONTROL framework containing three contrastive losses.

### 4.1 CONTRASTIVE LOSS ENHANCES CLASSIFICATION OF BCE LOSS

For BCE-based methods, it is difficult to identify same-class unseen pairs through only logit-level nearest-neighbor sample selection, since the nearest-neighbor sample may not necessarily belong to the same class, especially when the model is not yet well trained. Further, when the different-class unseen pairs are aligned by the BCE loss $\mathcal{L}_{\text{BCE}}$, the overall accuracy will be harmed.

Mathematically, given a nearest-neighbor unseen class pair $(x, v)$, we denote their marginal probability as $P_X$ and $P_V$, respectively. We can interpret the aligning procedure of pairs into a binary classification with label 1 when $(x, v)$ are positive same-class pairs sampled from the joint distribution $P_{XV}$, and with label 0 when $x$ and $v$ are independently sampled from the marginal distributions $P_X$ and $P_V$. As the nearest neighbor has some probability of not being the same class sample, we denote $\eta \in (0, 1)$ as the misclassification rate of the nearest-neighbor sample, and denote the joint probability of the nearest-neighbor pair $(x, v)$ as

$$P_{XV}^\eta = (1 - \eta)P_{XV} + \eta P_X P_V. \tag{3}$$

This joint probability can be interpreted as $x$ and $v$ having probability $1 - \eta$ to be the same-class samples and probability $\eta$ to be different-class ones.

Then for the BCE loss, we show that the different-class unseen pairs harm the classification of unseen-class samples. Denote $g(z; \theta')$ as the logits of feature embedding $z$. Then

$$
\begin{aligned}
\mathbb{E}_{P_{XV}^\eta} \mathcal{L}_{\text{BCE}} &= \mathbb{E}_{P_{XV}^\eta} - \log g(\phi(x))^\top g(\phi(v)) \\
&= (1 - \eta)\mathbb{E}_{(x,v) \sim P_{XV}} - \log g(\phi(x))^\top g(\phi(v)) + \eta \mathbb{E}_{x \sim P_X} \mathbb{E}_{v \sim P_V} - \log g(\phi(x))^\top g(\phi(v)) \\
&:= (1 - \eta)\mathbb{E}_{P_{XV}} \mathcal{L}_{\text{BCE}} + \eta M.
\end{aligned} \tag{4}
$$

Because $x \sim P_X$ and $v \sim P_V$ are independent, $g(\phi(x))$ and $g(\phi(v))$ are also independent, and therefore we have $g(\phi(x))^\top g(\phi(v)) = 0$ and thus $M \to -\infty$. This reveals a great gap between the BCE risk under distributions $P_{XV}^\eta$ and $P_{XV}$, indicating that the different-class nearest-neighbor pairs could severely harm the classification accuracy when using BCE loss.

On the other hand, contrastive losses are less affected. Specifically, we have

$$
\begin{aligned}
\mathbb{E}_{P_{XV}^\eta} \mathcal{L}_\phi &= \mathbb{E}_{P_{XV}^\eta} - \log \frac{\exp\left(\phi(x)^\top \cdot \phi(v^+)/\tau\right)}{\sum_{\phi(v^-) \in \mathcal{N}(\phi(x))} \exp\left(\phi(x)^\top \cdot \phi(v^-)/\tau\right)} \\
&= (1 - \eta)\mathbb{E}_{P_{XV}} \mathcal{L}_\phi + \eta \cdot \mathbb{E}_{x \sim P_X} \mathbb{E}_{v^+ \sim P_V} - \phi(x)^\top \cdot \phi(v^+)/\tau \\
&\quad + \eta \cdot \mathbb{E}_{x \sim P_X} \mathbb{E}_{v^- \sim P_V} \log \sum_{\phi(v^-) \in \mathcal{N}(x)} \exp\left(\phi(x)^\top \cdot \phi(v^-)/\tau\right) \\
&= (1 - \eta)\mathbb{E}_{P_{XV}} \mathcal{L}_\phi + \eta \cdot \log(|\mathcal{N}(x)|),
\end{aligned} \tag{5}
$$

where the last equation holds because $\phi(x)$, $\phi(v^+)$, and $\phi(v^-)$ are independent. Note that $\log(|\mathcal{N}(x)|)$ is a constant, and therefore $\mathbb{E}_{\mathrm{P}^\eta_{X_V}} \mathcal{L}_\phi$ and $\mathbb{E}_{\mathrm{P}_{X_V}} \mathcal{L}_\phi$ result in the same optimal classifier, indicating that contrastive losses are less affected by different-class nearest-neighbor pairs.

This motivates us to incorporate contrastive losses to realize the feature-level alignment and to enhance the classification of the BCE loss.

**Supervised Contrastive Loss.** Recall that in the BCE-based methods for open-world semi-supervised learning, the labels are correct and the labeled examples are aligned through same-class pairs. Similarly, for the labeled examples, we also align same-class pairs on the feature level by introducing the supervised contrastive learning loss (Khosla et al., 2020), which is well known for its ability to improve representation quality by aligning features within the same class. The form of supervised contrastive loss is

$$\mathcal{L}_{\text{SupSeen}} = -\frac{1}{|\mathcal{P}^S(x)|} \sum_{z^+ \in \mathcal{P}^S(x)} \log \frac{\exp\left(z^\top \cdot z^+ / \tau\right)}{\sum_{z^- \in \mathcal{N}(x)} \exp\left(z^\top \cdot z^- / \tau\right)}, \tag{6}$$

where the $\mathcal{P}^S(x)$ denotes the collection of same-class pairs. Note that $\mathcal{L}_{\text{SupSeen}}$ only applies to labeled data. After choosing an anchor, the positive set $\mathcal{S}(\mathbf{x})$ includes samples with the same label as the anchor, and the negative set $\mathcal{N}(x)$ includes samples from all classes.

**Nearest Neighbor Contrastive Loss.** Recall that the BCE loss is the key to classifying unseen classes for the BCE-based algorithms. For the unlabeled samples including both seen- and unseen-class samples, the pairs are selected as nearest-neighbor samples. As we discussed earlier, the falsely aligned different-class pairs could harm the classification performance of BCE loss, therefore we turn to incorporate the lessly affected nearest neighbor contrastive loss $\mathcal{L}_{\text{SupNN}}$ to achieve feature-level alignment. The specific form is

$$\mathcal{L}_{\text{SupNN}} = -\frac{1}{|\mathcal{P}^N(x)|} \sum_{z^+ \in \mathcal{P}^N(x)} \log \frac{\exp\left(z^\top \cdot z^+ / \tau\right)}{\sum_{z^- \in \mathcal{N}(x)} \exp\left(z^\top \cdot z^- / \tau\right)}, \tag{7}$$

where the $\mathcal{P}^N(x)$ denotes the collection of nearest-neighbor pairs. Note that $\mathcal{L}_{\text{SupNN}}$ is a feature-level counterpart for the BCE loss. It applies to both labeled data and unlabeled data. After choosing an anchor, the positive set $\mathcal{P}^N(\mathbf{x})$ includes the nearest neighbor of the anchor, and the negative set $\mathcal{N}(\mathbf{x})$ includes all the other samples.

$\mathcal{L}_{\text{SupNN}}$ allows similar sample pairs to have more similar feature representations, which naturally facilitates BCE loss to learn more similar logits, and in turn increase the proportion of same-class pairs in all nearest-neighbor pairs. As a result, $\mathcal{L}_{\text{SupNN}}$ guides the model for better seen/unseen classification.

## 4.2 Uniformity in Contrastive Loss Avoids Unseen Class Collapse

It is easy to see that a trivial solution to the BCE loss on the unseen classes is to classify all samples into one class, whereas this will severely harm the classification accuracy on the unseen classes too. Despite the introduction of the entropy loss $\mathcal{L}_{\text{Entropy}}$, the distinguishment among unseen classes still remains to be a great challenge.

Aside from providing feature-level alignment, we demonstrate that contrastive losses naturally avoid logit collapse of unseen classes. Specifically, by Wang & Isola (2020), the contrastive loss can be decomposed into an alignment term and a uniformity term, i.e.

$$\lim_{|\mathcal{N}(x)| \to \infty} \mathbb{E}\, \mathcal{L}_\phi(x; \tau, \mathcal{P}(x), \mathcal{N}(x)) - \log K = -\mathbb{E}_{z, z^+ \in \mathcal{P}(\mathbf{x})} \exp(z^\top \cdot z^+ / \tau) \tag{8}$$

$$+ \mathbb{E}_{z \in \mathcal{P}(\mathbf{x})} \log \mathbb{E}_{z^- \in \mathcal{N}(\mathbf{x})} \exp(z^\top \cdot z^- / \tau), \tag{9}$$

where equation 8 represents the alignment term that aligns the feature representations of the positive pairs, and equation 9 represents the uniformity term which forces the representations to be uniformity distributed throughout the entire feature embedding space and thus avoids feature collapse.

Naturally, as both logit-level and feature-level alignment is achieved, $g(\cdot)$ maintains the spatial structure between feature representations and logits. Therefore, by avoiding feature-level collapse, the contrastive losses also avoid logit-level collapse of the unseen classes.

### 4.3 UNSUPERVISED CONTRASTIVE LOSS SERVES AS CONSISTENCY REGULARIZATION

Moreover, recall that for the open-world semi-supervised learning problem, consistency regularization is often used to further boost the classification performance, where logit-level consistency regularization aligns the logits of two augmented views. Similarly, as data augmentation is widely used and well-studied in contrastive learning, we naturally introduce unsupervised contrastive loss to achieve feature-level consistency regularization. The specific form is

$$\mathcal{L}_{\text{SimAll}} = -\frac{1}{|\mathcal{P}^A(x)|} \sum_{z^+ \in \mathcal{P}^A(x)} \log \frac{\exp\left(z^\top \cdot z^+/\tau\right)}{\sum_{z^- \in \mathcal{N}(x)} \exp\left(z^\top \cdot z^-/\tau\right)}, \tag{10}$$

where the $\mathcal{P}^A(x)$ denotes the collection of positive pairs augmented from the same natural sample. $\mathcal{L}_{\text{SimAll}}$ applies to both labeled data and unlabeled data which is beneficial to classifying both seen classes and unseen classes.

### 4.4 OVERALL

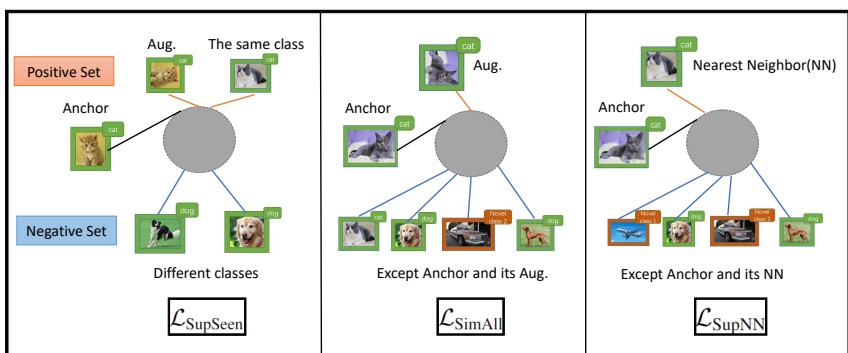

Figure 4: The framework of our proposed CONTROL.

In conclusion, our proposed CONTROL consists of a replaceable open-world semi-supervised learning module and a simple but efficient contrastive learning module as shown in Figure 4. For different open-world semi-supervised learning algorithms, we will all use $\mathcal{L}_{\text{OWSSL}}$ to represent the original loss. To strengthen the open-world semi-supervised learning algorithm, our proposed CONTROL contains supervised contrastive learning part $\mathcal{L}_{\text{SupSeen}}$, unsupervised contrastive learning part $\mathcal{L}_{\text{SimAll}}$ and nearest neighbor contrastive learning part $\mathcal{L}_{\text{SupNN}}$ which are iteratively optimized together with $\mathcal{L}_{\text{OWSSL}}$ during the training process:

$$\mathcal{L}_{\text{CONTROL}} = \mathcal{L}_{\text{OWSSL}} + \lambda_1 \mathcal{L}_{\text{SupSeen}} + \lambda_2 \mathcal{L}_{\text{SimAll}} + \lambda_3 \mathcal{L}_{\text{SupNN}}, \tag{11}$$

where $\lambda_1$, $\lambda_2$ and $\lambda_3$ in the formula are trade-off parameters.

## 5 EXPERIMENTS

### 5.1 SETUP

**Datasets.** We evaluate our framework CONTROL combined with representative open-world semi-supervised learning algorithms on common SSL datasets: CIFAR-10, CIFAR-100 (Krizhevsky & Hinton, 2009) and ImageNet-100. On CIFAR-10 we choose 5 classes as seen classes and the rest as unseen classes. On CIFAR-100 and ImageNet-100 we choose 50 classes as seen classes and the rest as unseen classes. Following the same principles as NACH and ORCA, we select 50% samples of the seen classes as labeled data, i.e. the unlabeled dataset is a 1:2 mix of samples from seen classes and unseen classes.

**Baselines.** We choose NACH and ORCA as our baseline methods and use these baseline methods to demonstrate the effectiveness of our framework.

Table 1: Classification accuracy of compared methods on seen, unseen and all classes on CIFAR-10 and CIFAR-100 (Krizhevsky & Hinton, 2009). We use the hungarian algorithm (Kuhn, 1955) to evaluate unseen classes accuracy and all classes accuracy, and seen classes accuracy is evaluated directly by whether the labeled data are divided into the right classes. For a fairer comparison, all experiments use the same backbone ResNet18 (He et al., 2016) as (Cao et al., 2022). Following (Cao et al., 2022), we also show the results of traditional semi-supervised learning method Fixmatch (Sohn et al., 2020), the open set semi-supervised algorithms DS3L (Guo et al., 2020), CGDL (Sun et al., 2020), and the novel class discovering algorithms DTC (Han et al., 2019)and Rankstats (Han et al., 2020). In particular, we evaluate our proposed framework by combining ORCA (Cao et al., 2022) and NACH (Guo et al., 2022) with the backbone fully unlocked, denoted as ORCA* and NACH*.

| Method | CIFAR-10 | | | CIFAR-100 | | |
|---|---|---|---|---|---|---|
| | Seen | Unseen | all | Seen | Unseen | all |
| Fixmatch | 71.5 | 50.4 | 49.5 | 39.6 | 23.5 | 20.3 |
| DS3L | 77.6 | 45.3 | 40.2 | 55.1 | 23.7 | 24.0 |
| CGDL | 72.3 | 44.6 | 39.7 | 49.3 | 22.5 | 23.5 |
| DTC | 53.9 | 39.5 | 39.3 | 31.3 | 22.9 | 18.3 |
| Rankstats | 86.6 | 81.0 | 82.9 | 36.4 | 28.4 | 23.1 |
| OpenCon | 87.1 | 90.7 | 89.4 | 70.4 | 50.2 | 55.5 |
| ORCA | 88.2 | 90.4 | 89.7 | 66.9 | 43.0 | 48.1 |
| ORCA + CONTROL | **88.4** | **91.9** | **90.7** | 67.6 | **44.8** | **49.6** |
| ORCA* | 89.0 | 90.0 | 89.3 | 67.0 | 42.2 | 48.8 |
| **ORCA* + CONTROL** | **90.6** | **90.6** | **90.4** | 69.5 | **47.2** | **53.2** |
| NACH | 89.5 | 92.2 | 91.3 | 68.7 | 47.0 | 52.1 |
| **NACH + CONTROL** | **89.9** | **93.4** | **92.2** | 70.1 | **48.5** | **53.0** |
| NACH* | 93.3 | 94.1 | 94.1 | 73.5 | 50.1 | 56.8 |
| **NACH* + CONTROL** | **93.5** | **95.0** | **94.3** | **74.1** | **53.3** | **58.0** |

Table 2: Classification results on ImageNet-100 (Russakovsky et al., 2015). All experimental results are the average of three runs. + denotes the combination of CONTROL. For a fair comparison, all experiments use the same backbone ResNet50 as (Cao et al., 2022).

| Method | Fixmatch | DS3L | Rankstats | OpenCon | ORCA | ORCA+ | NACH | NACH+ |
|---|---|---|---|---|---|---|---|---|
| Seen | 65.8 | 71.2 | 47.3 | 90.4 | 88.7 | 89.4 | 90.4 | **90.9** |
| Unseen | 36.7 | 32.5 | 28.7 | 80.6 | 72.3 | 77.8 | 75.6 | **81.8** |
| All | 34.9 | 30.8 | 40.3 | 83.2 | 77.4 | 81.0 | 79.6 | **84.0** |

## 5.2 MAIN RESULTS

The classification mean accuracy over three runs on CIFAR-10 and CIFAR-100 is in Table 1. The results show that our proposed method is an effective framework for open-world semi-supervised learning algorithms. We take CIFAR100 for example, on seen classes classification, we achieved 3.7% improvement on ORCA and 0.8% improvement on NACH. On unseen class classification, we achieved 11.8% improvement on ORCA and 6.4% improvement on NACH. On all classes classification, we also achieved 9% improvement on ORCA and 2.1% improvement on NACH. The performance improvement for unseen classes mentioned above mainly stems from the enhancement of BCE loss in our framework, while the performance improvement for all class classifications primarily comes from the uniformity component in CONTROL, which further mitigates the collapse of existing methods.

The results also show that compared with OpenCon, we achieved 5.2% improvement on seen classes, 2.2% improvement on unseen classes, and 4.5% improvement on all classes. These performance improvements show that compared with the contrastive learning method only, the semi-supervised learning part combined with CONTROL can also achieve the best results. As shown in Table 2, we also conducted our experiments on a more realistic dataset ImageNet-100 (Russakovsky et al., 2015), and we achieved consistent performance improvement with CONTROL as mentioned above.

Table 3: Ablation Studies: Seen, Unseen and All classes classification accuracy for $\mathcal{L}_{\mathrm{SupSeen}}$, $\mathcal{L}_{\mathrm{SimAll}}$, $\mathcal{L}_{\mathrm{SupNN}}$ on CIFAR-100 combined with NACH.

| $\mathcal{L}_{\mathbf{SupSeen}}$ | $\mathcal{L}_{\mathbf{SimAll}}$ | $\mathcal{L}_{\mathbf{SupNN}}$ | **Seen** | **Unseen** | **All** |
|---|---|---|---|---|---|
| W.O. | W.O. | W.O. | 73.5 | 50.1 | 56.8 |
| W. | W.O. | W.O. | 74.0 | 50.2 | 56.9 |
| W.O. | W. | W.O. | 73.6 | 50.7 | 57.0 |
| W.O. | W.O. | W. | 74.0 | 51.4 | 56.9 |
| W. | W. | W. | 74.1 | 53.3 | 58.0 |

## 5.3 ABLATION STUDIES

**Analysis on $\mathcal{L}_{\mathbf{SupSeen}}$ and $\mathcal{L}_{\mathbf{SimAll}}$.** As we analyzed before, $\mathcal{L}_{\mathrm{SupSeen}}$ can align features within the same classes which can be regarded as an enhancement of BCE loss and $\mathcal{L}_{\mathrm{SimAll}}$ can be regarded as a form of consistency regularization that enables the model to learn a better representation from the sample level. Therefore, as shown in Table 3, combined with $\mathcal{L}_{\mathrm{SupSeen}}$ and $\mathcal{L}_{\mathrm{SimAll}}$, we achieved 0.2% improvement on seen classes, 2.7% improvement on unseen classes and 1.1% improvement on all classes. This demonstrates the effectiveness of our CONTROL framework in both avoiding unseen class collapse and enhancing classification accuracy through additional consistency regularization.

**Analysis on $\mathcal{L}_{\mathbf{SupNN}}$.** As shown in Table 3, combined with $\mathcal{L}_{\mathrm{SupNN}}$, we achieved 0.7% improvement on seen classes, 2.6% improvement on unseen classes and 0.2% improvement on all classes. The experimental results are consistent with our previous theoretical analysis. In CONTROL, the utilization of $\mathcal{L}_{\mathrm{SupNN}}$ allows similar sample pairs to have more similar representations at the feature level, and BCE loss will learn more similar logits to better guide the model for unseen classes classification.

## 5.4 FURTHER ANALYSIS

Based on our theoretical analysis, we will use the following experiments to further illustrate why our proposed framework is useful for improving the classification accuracy of unseen classes based on BCE loss. In Table 4, we show the ratio of unseen class prediction and the ratio of unseen-unseen pairs to figure out why our proposed framework is useful. For the ratio of unseen classes prediction, we take the sample of unseen classes whose prediction is unseen classes as the numerator, we take the total number of unseen classes samples as the denominator. A larger ratio means that the algorithm is better at classifying unseen classes since our goal is to have less confusion between seen classes and unseen classes. The results show that, after using our framework, we gained 2.77% improvement. For the ratio of unseen-unseen pairs, we take the sum of unseen class samples whose nearest neighbors are also unseen class samples as the numerator, we take the total number of unseen class samples as the denominator. A larger ratio means that the algorithm is better at classifying unseen classes since the larger it is, the less seen-unseen pairs happen. The results show that, after using our framework, we gained 2.35% improvement.

Table 4: Ratio of unseen classes prediction and Ratio of unseen-unseen pairs for NACH with or without CONTROL. All experimental results are the average of three runs on CIFAR-100.

| Method | Ratio of unseen class prediction | Ratio of unseen-unseen pairs |
|---|---|---|
| NACH | 79.37 | 87.77 |
| **NACH + CONTROL** | **82.14** | **90.12** |

## 6 CONCLUSION

We propose a simple and efficient contrastive learning framework CONTROL, which can be easily adapted to open-world semi-supervised learning algorithms. The importance of our framework in improving the unseen classification performance of open-world semi-supervised learning algorithms has been validated theoretically. Additionally, our proposed framework can further mitigate the risk of representation collapse. We have also designed experiments to demonstrate why CONTROL is effective in classifying unseen classes.

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
