# OpenReview forum: "CONTROL: A Contrastive Learning Framework for Open World Semi-Supervised Learning"
_ICLR.cc/2024/Conference — Submitted to ICLR 2024_

### Official Review · Reviewer_mngU · 2023-10-23

**Soundness:** 1 poor
**Presentation:** 1 poor
**Contribution:** 1 poor
**Rating:** 5
**Confidence:** 4

**Summary:**

This paper enhances previous open-world semi-supervised learning (OWSSL) methods such as ORCA and NACH by adding various contrastive learning losses, including supervised (SupCon), nearest neighbor (SupNN), and self-supervised (SelfCon) ones.

**Strengths:**

- Open-world semi-supervised learning is a practical problem.
- The proposed method improves previous approaches, such as ORCA and NACH.

**Weaknesses:**

**Unclear presentation and unjustified claims**

Before delving into the other concerns, the presentation lacks clarity and is missing essential information.

- In Sec. 1 - paragraph 3, the paper categorizes OWSSL methods into "BCE-based" and "Other" methods. Why is the use of BCE loss a meaningful categorization?
- In Sec. 1 - paragraph 4, the paper claims that using MSE instead of BCE lacks "flexibility." What is the definition of flexibility in this context, and why is BCE considered better than MSE for this purpose?
- In Sec. 1 - paragraph 5, the paper suddenly discusses "seen-unseen pairs." Why is this concept important, and how is it related to the problem? This relationship is not explained.
- In Sec. 2 - paragraph 2, the paper claims that previous works do not consider optimization at the feature level. However, they also jointly optimize the feature and classifier.
- In Sec. 2 - paragraph 3, the paper claims that OpenCon is incapable of sustaining "continuous optimization" of representations. What is continuous optimization, and why can't OpenCon achieve this?
- In Sec. 3.2, why should one consider BCE-based methods? BCE-based methods are just one specific approach by Cao et al. and some follow-ups. Eq. (1) does not represent all OWSSL methods.
- ...but not limited to.

Aside from the unclear overall logic, the sentences are significantly unpolished. Many grammatical errors make it difficult to grasp the meaning.
There are also typos and inconsistent use of terminologies. For example, "L" in the abstract, uncapitalized "all" in Table 1, unformatted "seen/unseen" in Sec. 3.1, and "OWSSL" is not defined in Sec. 3.2.


---
**Proposed method is merely a combination of existing techniques**


The motivation if the method is (1) enhance the separability of known class features and (2) prevent the collapse of unknown class features.
The paper employs SupCon (and a NN variant) for (1) and SelfCon for (2).

There are two issues here:
1. It is obvious that contrastive learning would enhance performance. In fact, this benefit for OWSSL has already been demonstrated in prior works, such as GCD [1] and OpenCon [2].
2. The addition of Sup/SelfCon generally enhances the models, not only for the OWSSL problem. To verify this, one can apply the proposed method to (1) the standard SSL setup and (2) other SSL methods. I believe it will also result in improvements in those cases, suggesting that the method does not exclusively target the OWSSL issue.

---
**Missing important baselines**

The paper omits a comparison with a highly related work, OpenCon [2]. The reasons provided in the paper for avoiding this comparison are not convincing. Additionally, GCD [1] is highly relevant to the paper but is missing from the comparison.
Furthermore, the paper does not include a comparison with papers published in 2023, such as RoPAWS [3], which also employs contrastive learning for SSL. Not only them, but there is also a long list of related works.
The OpenCon paper has 6 pages of references, in contrast to this paper with only 2 pages. Checking them would enable a more comprehensive context in the related work section, thereby making the paper more self-contained.

[1] Generalized Category Discovery, CVPR'22.\
[2] OpenCon: Open-world Contrastive Learning. TMLR'22.\
[3] RoPAWS: Robust Semi-supervised Representation Learning from Uncurated Data, ICLR'23.

**Questions:**

1. Why is BCE better than other approaches, such as MSE?
2. Why is OpenCon not capable of continuous optimization?
3. Does the proposed method also improve non-OW SSL benchmarks and other OWSSL methods?

---

> ### Author Response · Authors · 2023-11-22
>
> ## Thanks to your valuable feedback. We sincerely appreciate your constructive comments.
>
> ## R 4.1 In Sec. 1 - paragraph 3,  why is the use of BCE loss a meaningful categorization?
>
> We figure out that the similarity function is honed by directly computing the similarity of all sample pairs and their logits counterparts in other methods. However, not all the similarity of sample pairs can  efficiently guide the subsequent learning process since unseen class samples are prone to be mislabeled as seen classes in initial training phases. That's why BCE-based is important and could be regarded as a meaningful categorization.
>
> ## R 4.2 paragraph 4, the paper claims that using MSE instead of BCE lacks "flexibility." ...
>
> In the original version we did not elucidate this issue clearly, in fact, it is the difference in the way sample pairs are selected that defines the different categories of methods, and the BCE-based methods are based on the nearest-neighbor feature selection which is more flexible. So we say that the BCE-based methods are flexible, not that the BCE is superior to the MSE.
>
> ## R 4.3 In Sec. 1 - In Sec. 1 - paragraph 5, the paper suddenly discusses "seen-unseen pairs." Why is this concept important, and how is it related to the problem? This relationship is not explained.
>
> Because these methods utilizes pairwise clustering in order to give the model the ability to classify seen and unseen classes, the quality of the sample pairs is very important, so we have to pay attention to the seen-unseen sample pairs.
>
> ## R 4.4 In Sec. 2 - paragraph 2, the paper claims that previous works do not consider optimization at the feature level. However, they also jointly optimize the feature and classifier.
>
> You may refer to our answer to reviewer fiFv  R 3.1.
>
> ## R 4.5 In Sec. 2 - paragraph 3, the paper claims that OpenCon is incapable of sustaining "continuous optimization" of representations. What is continuous optimization, and why can't OpenCon achieve this?
>
> We give a detailed explanation in our latest PDF.
>
> ## R 4.6 In Sec. 3.2, why should one consider BCE-based methods? BCE-based methods are just one specific approach by Cao et al. and some follow-ups. Eq. (1) does not represent all OWSSL methods.
>
> In open world semi-supervised learning problem, high quality chosen samples are utilized to give the model the ability to classify both seen and unseen classes. The strategy for selecting the samples is critical. And BCE-based methods are more suitable for choosing high quality samples since these methods can choose samples pairs in a more flexible way as explained before.
>
> ## R 4.6  Aside from the unclear overall logic, the sentences are significantly unpolished. Many grammatical errors make it difficult to grasp the meaning. There are also typos and inconsistent use of terminologies. For example, "L" in the abstract, uncapitalized "all" in Table 1, unformatted "seen/unseen" in Sec. 3.1, and "OWSSL" is not defined in Sec. 3.2.
>
> We have made the appropriate changes in the paper.
>
> ## R 4.7 The motivation if the method is (1) enhance the separability of known class features and (2) prevent the collapse of unknown class features. The paper employs SupCon (and a NN variant) for (1) and SelfCon for (2).
>
> You may refer to our answer to reviewer fiFv  R 3.1.
>
> ## R 4.8 Missing important baselines such as GCD , OpenCon and RoPAWS.
>
> We have compared Opencon in Table 1, where the combination of NACH and control outperforms Opencon.
> we have added GCD to the comparison methods.
> RoPAWS is not an OWSSL algorithm, and its goal is not to classify both seen and unseen classes, but rather to better ensure the correctness of seen categories when unlabeled data has unseen classes, so we will not discuss it.
>
> The results of additional experiments are as follows， we evaluate all the methods on CIFAR-100 dataset.
> |  CIFAR100  |seen acc  |unseen acc  |all acc |
> |----  |----  |----  |----  |
> | ORCA  | 66.9 |43.0 |48.1	|
> | ORCA+CONTROL  | 67.6 |44.8 |49.6	|
> | Opencon |69.3 |47.6 |51.8	|
> | GCD  | 69.8 |42.9 |46.5	|
> | NACH  | 68.7 |47.0 |52.1	|
> | NACH+CONTORL | 70.1 |48.5 |53.0	|
>
> We can see that, the combination of NACH and our proposed CONTROL still gives the best results among all the compared methods.
>
>
> ## R 4.9 Does the proposed method also improve non-OW SSL benchmarks and other OWSSL methods?
>
> We illustrate this with Fixmatch, a representative deep semi-supervised learning algorithm with 250 labeled data on cifar10 dataset using wideresnet-28-2 as backbone, in conjunction with L_simALL and L_supSeen in our proposed framework, with the following results:
> |      |original  |with CONTROL |
> |----  |----  |----  |
> |ACC   |95.13      |95.28  |
>
> We can see that even doing feature-level optimization based on the semi-supervised learning algorithm in the traditional setup results in a performance gain, but they are far less than in OWSS setup, which demonstrates the reasonableness of our proposed combination of CONTROL and OWSSL algorithms.

---

> ### Comment · Reviewer_mngU · 2023-12-04
> **Response to the Rebuttal**
>
> After discussing with other reviewers, I think that the rebuttal did not fully address the initial concerns:
> - Presentation: The overall research insight is still not effectively conveyed in the manuscript. A major revision may be necessary to improve clarity.
> - Novelty: If I understand correctly, the rebuttal claims that the technical contribution is in the analysis of previous approaches, and the proposed solution (SupCon, NN) straightforwardly arises from this analysis. This may also relate to Reviewer 1Kmz's narration concern; the delivery of this claim could be clarified further.
> - Experiments: I believe the experimental results are okay, although combining additional objectives should enhance performance in general. Nevertheless, the combination of OpenCon and CONTROL in R2.2 is not convincing enough.

---

### Official Review · Reviewer_fiFv · 2023-11-01

**Soundness:** 1 poor
**Presentation:** 2 fair
**Contribution:** 1 poor
**Rating:** 5
**Confidence:** 4

**Summary:**

This paper introduces a contrastive loss designed to enhance the performance of open world semi-supervised learning (OWSSL). Specifically, it proposes three distinct loss functions tailored for nearest neighbor contrastive learning, supervised contrastive learning, and unsupervised contrastive learning. The primary aim of these functions is to address and mitigate the representation collapse issue. The method presented has achieved state-of-the-art performance on benchmark datasets including CIFAR-10, CIFAR-100, and ImageNet-100.

**Strengths:**

1. The proposed approach is straightforward and easy to understand.
2. The method's efficacy has been demonstrated across various datasets.

**Weaknesses:**

**1. Novelty Concerns**:
The proposed method essentially adds three types of contrastive losses to the existing OWSSL loss. Of these, the supervised contrastive loss is merely a standard contrastive loss. Additionally, both the NN contrastive loss and the SimAll contrastive loss that utilizes augmented images as positive keys lack distinctive features compared to existing research [A, B].

[A] With a Little Help from My Friends: Nearest-Neighbor Contrastive Learning of Visual Representations, ICCV 2021. \
[B] Rethinking the Augmentation Module in Contrastive Learning: Learning Hierarchical Augmentation Invariance with Expanded Views, CVPR 2022.

**2. Overall Paper Completeness**:
The completeness of the paper is generally lacking. The proof provided in Section 4.1 for motivation is not rigorous, with some intermediate steps omitted. And there are three hyperparameters related to the loss. However, the paper does not provide details on the values chosen for these parameters, nor does it discuss their sensitivity or robustness through experiments.

**3. Reproducibility Concerns**:
The absence of a reproducibility statement raises doubts about the possibility of replicating the claimed state-of-the-art performance.

**Questions:**

In Section 3.1, shouldn't $C_{\text{seen}}$ be defined as $C_L$ instead of $C_{\text{seen}} = C_L \cap C_U$?

---

> ### Author Response · Authors · 2023-11-22
>
> ## Thanks to your valuable feedback. We sincerely appreciate your constructive comments.
>
> ## R 3.1 Novelty Concerns: The proposed method essentially adds three types of contrastive losses to the existing OWSSL loss. Of these, the supervised contrastive loss is merely a standard contrastive loss. Additionally, both the NN contrastive loss and the SimAll contrastive loss that utilizes augmented images as positive keys lack distinctive features compared to existing research.
>
> Existing methods (e.g., BCE based methods) consider the open world semi-supervised learning problem from the logits level optimization, but we point out through theoretical analysis that when there is the phenomenon of seen-unseen pairs, the optimal classifiers learned through logits optimization are not the same as the ideal state (see Eqn. 4), but optimization based on comparative learning at the feature level ensures that even in the case of mismatches, we can still theoretically show that the classifiers we learn are the same as the ideal case (see Eqn. 5). We point out this key issue and validate our viewpoint with extensive experiments. To the best of our knowledge, no theoretical proof of the importance of feature-level optimization for OWSSL has been proposed and given before. All the three losses are proposed in the feature-level optimization which have theoretically guaranteed to get a performance gain.
>
> ## R 3.2 Overall Paper Completeness: The completeness of the paper is generally lacking. The proof provided in Section 4.1 for motivation is not rigorous, with some intermediate steps omitted. And there are three hyperparameters related to the loss. However, the paper does not provide details on the values chosen for these parameters, nor does it discuss their sensitivity or robustness through experiments.
>
> About the proof provided in Section 4.1, we illustrate the importance of feature-level optimization by whether the optimal classifier learned in the presence of mispairing is consistent with the ideal optimal classifier, Eq. 3 illustrates inconsistency at logits level and Eq. 4 illustrates consistency at feature level.
>
> As mentioned before all the three losses are proposed in the feature-level optimization which have theoretically guaranteed to get a performance gain.
>
> About the choice of parameters, the results of additional stripping experiments are as follows， we evaluate our methods on CIFAR-100 dataset with different parameters.
> |  lambda1   | lambda2  |lambda3  |seen acc  |unseen acc  |all acc |
> |  ----  | ----  |----  |----  |----  |----  |
> | 0.1  | 0.1 |0.1 |74.0	|47.9	|54.7|
> | 0.1  | 0.1 |0.3 |74.1	|49.4|	55.2|
> | 0.1  | 0.1 |0.5 |74.1	|50.8|	56.5|
> | 0.3  | 0.1 |0.1 |74.2	|51.1|	57.0|
> | 0.3  | 0.1 |0.3 |74.1	|50.6|	57.8|
> | 0.3  | 0.1 |0.5 |74.2	|49.5|	55.4|
> | 0.5  | 0.1 |0.1 |74.0	|48.1|	54.9|
> | 0.5  | 0.1 |0.3 |74.1 |53.3|	58.0|
> | 0.5  | 0.1 |0.5 |74.0	|53.1|	58.0|
>
> The values 0.5 0.1 0.3 that we report in the paper are optimal for the parameter combinations.
>
> ## R 3.3 Reproducibility Concerns: The absence of a reproducibility statement raises doubts about the possibility of replicating the claimed state-of-the-art performance.
>
> All the results are reproducible, and we'll make our code publicly available upon acceptance.
>
> ## R 3.4 In Section 3.1, shouldn't C_seen be defined as C_L instead of C_seen = C_L cap C_U
>
> Yes, the two are equivalent in our setup, and we've made changes for better understanding by the reader.

---

> > ### Comment · Reviewer_fiFv · 2023-12-05
> >
> > Thank you for the effort put into preparing the rebuttal. Having reviewed the response and considered the feedback from other reviewers, I have decided to increase my rating from 3 to 5. However, I maintain my position towards rejection. My concerns primarily revolve around the novelty of the work. Additionally, it appears that the proposed method demonstrates considerable sensitivity to hyperparameter settings, which is another point of concern for me.

---

### Official Review · Reviewer_1Kmz · 2023-11-07

**Soundness:** 3 good
**Presentation:** 2 fair
**Contribution:** 3 good
**Rating:** 5
**Confidence:** 4

**Summary:**

The paper proposes to apply contrastive learning for an interesting task setting, open-world semi-supervised learning. Though there has been previous work that attempts to introduce contrastive learning for semi-supervised learning (i.e., OpenCon), the paper claims that previous work does not consider the open-world task setting, which leads to failure due to distribution shift and unseen classes. The paper proposes to adapt various loss terms used in other SSL tasks. In addition, the paper focuses on performing SSL on the feature-level representations, where the author provides theoretical analysis to argue the superiority of contrastive loss over conventional BCE loss. Experiments combine the proposed CONTROL technique with two existing baselines and show noticeable improvements in the accuracy of both seen and unseen classes.

**Strengths:**

- The paper investigates a somewhat interesting task setting, termed open-world SSL, which is of interest for the deployment of vision models in the wild.
- The presentation for the benefit of using the contrastive loss over conventional BCE loss is good with theoretical analysis, which provides a good motivation for introducing contrastive loss to open-world SSL, and can potentially facilitate future research.
- The paper provides good experimental results where the proposed method is combined with two previous baselines on CIFAR-10, CIFAR-100, and the ImageNet-100 datasets. The ablation study of CONTROL on CIFAR-100 also justifies the motivation for different introduced loss terms. The final analysis on the improved 'recall' rate of samples from unseen classes also provides good motivation for the proposed method

**Weaknesses:**

- The main weakness of the paper lies in the narration of the paper, which makes it hard for a broader audience to capture the contributions of the paper. In many parts of the paper, the writing is focused on *what* is done but not *why* this is better than alternative methods. This makes the paper seem like a technical report rather than a research paper. A good paper is expected to provide insights to other researchers on 'why' the method works and how it can help with future research and real-world applications. This issue is particularly concerning in the introduction section, which is supposed to provide readers with high-level motivation of the methods. However, in the introduction, the third paragraph lists BCE-based methods and non-BCE methods without discussing when each of the methods is preferred and why each is motivated. Though the fourth paragraph attempts to argue for the superiority of BCE-based methods for open-world SSL, it again does not talk about high-level motivation for BCE-based methods. Instead, it directly dives into the details of similarity functions and talks about 'pulling closer sample pairs'. Such a discussion misses out a lot of context and causes confusion on what are the actual contribution of the paper. Also, the highlighted question in the introduction section seems to overlap greatly with the contribution with OpenCon. The only difference is that the authors claim CONTROl is 'unified' and 'open-world', but there is no further analysis on these two factors and how CONTROL differentiates itself from OpenCon.
- The paper seems to miss out on the addition of CONTROL to an important baseline OpenCon. OpenCon also proposes to apply contrastive learning for SSL. The paper argues that OpenCon is 'incapable of sustaining continuous optimization of representations during the process of semi-supervised learning'. However, from the final table 1 and table 2, the performance of OpenCon is pretty close to NACH+CONTROL, which somewhat undermines the claim of the novelty and contributions of CONTROL. A more convincing experiment would be to extend OpenCon with techniques proposed in this paper.

**Questions:**

I am slightly leaning toward the negative side mainly due to the concerns in narration and minor concerns in the experiment, as listed in the weaknesses above. However, I am open to discussions if the authors could kindly address the following questions

- **Narration.** Can the author elaborate more on high-level discussion on the motivation of the method and how it differentiates from previous work? This can be regarded as a revision of the introduction section but can be brief just for discussion now.
- **A small-scale additional experiment.** It would greatly enhance the claim of contributions in this paper if the author could provide a small-scale experiment on a dataset where CONTROl is combined with OpenCon.

---

> ### Author Response · Authors · 2023-11-22
>
> ## Thanks to your valuable feedback. We sincerely appreciate your constructive comments.
>
> ## R2.1 Narration. Can the author elaborate more on high-level discussion on the motivation of the method and how it differentiates from previous work? This can be regarded as a revision of the introduction section but can be brief just for discussion now.
>
> We have rewritten the introduction as you suggested.
>
> Existing methods (e.g., BCE based methods) consider the open world semi-supervised learning problem from the logits level optimization, but we point out through theoretical analysis that when there is the phenomenon of seen-unseen pairs, the optimal classifiers learned through logits optimization are not the same as the ideal state (see Eqn. 4), but optimization based on comparative learning at the feature level ensures that even in the case of mismatches, we can still theoretically show that the classifiers we learn are the same as the ideal case (see Eqn. 5).
>
> ## R2.2 A small-scale additional experiment. It would greatly enhance the claim of contributions in this paper if the author could provide a small-scale experiment on a dataset where CONTROl is combined with OpenCon.
>
>
> We test the performance of CONTROL on the CIFAR100 dataset using the opencon training results as a backbone and not using the opencon training results as a backbone, respectively.
>
> | CIFAR100 |Seen ACC |Unseen ACC| All ACC|
> |----  |----  |----  | ----  |
> |CONTROL without OPENCON backbone  |70.1      |48.5 |53.0  |
> |CONTROL with OPENCON backbone  |41.5      |22.6  |24.9  |
>
> As we can see, OPENCON achieves good results, but as a backbone it does not further improve performance when combined with our proposed framework. We can see the advantages of combining CONTORL with existing semi-supervised learning algorithms from this perspective.

---

### Official Review · Reviewer_1XMG · 2023-11-09

**Soundness:** 2 fair
**Presentation:** 2 fair
**Contribution:** 3 good
**Rating:** 5
**Confidence:** 2

**Summary:**

The paper presents CONTROL, a contrastive learning framework designed to improve the performance of open-world semi-supervised learning (SSL) algorithms. CONTROL addresses the presence of unseen classes by integrating three optimization objectives: nearest-neighbor contrastive learning, supervised contrastive learning, and unsupervised contrastive learning. Experimental evaluations on CIFAR and ImageNet show promising improvements over state-of-the-art methods.

**Strengths:**

+ The proposed three objectives are interesting and helpful.
+ The proof and derivation of the objectives are comprehensive and easy to understand.
+ Extensive evaluations and comparisons to baselines show promising results.

**Weaknesses:**

The writing of this paper is kind of informal, some examples include:
- The whole Section 3 missed a lot of details, lacking the basic problem setting, and key components of relied models, which makes the whole method hard to understand.
- Fig 1 and Fig 4 should be further polished. There are also overlaps between those two figures.

Next, It's hard for me to evaluate the contribution of the three loss functions, as all of them have been somehow explored in other areas before, and open-world semi-supervised learning is a new task. I have no sense if the contribution is huge or minor, so as the performance gain.

**Questions:**

- Page 2: what is a seen-unseen pair, Figure 2 is also hard to understand.

- It should be good to include some real-world results except the two datasets.

- Will the additional objectives significantly increase the training time, regarding the speed for convergence?

- For Table 3, why the $L_{superseen}$ and $L_{SimAll}$ are bundled together? I didn't see the dependency.

I rate a borderline for now and will take the rebuttal and opinions from other reviews into consideration.

----

After seeing the rebuttal and comments from the other reviewers, I tend to maintain my original rating.

---

> ### Author Response · Authors · 2023-11-22
>
> ## Thanks for your valuable feedback. We sincerely appreciate your constructive comments.
>
> ## R 1.1 The whole Section 3 missed a lot of details, lacking the basic problem setting, and key components of relied models, which makes the whole method hard to understand.
>
> We have reorganized section 3 according to your suggestions.
>
> ## R 1.2 Fig 1 and Fig 4 should be further polished. There are also overlaps between those two figures.
> Figure 4 uses the same example as Figure 1, and we do so to better illustrate whether each part of our proposed loss function should correspond to labeled or unlabeled data.
>
> ## R 1.3 Page 2: what is a seen-unseen pair, Figure 2 is also hard to understand.
>
> In the new version we have further elaborated seen-unseen pairs.
>
> ## R 1.4 It should be good to include some real-world results except the two datasets.
>
> Imagenet is widely adopted in SSL studies, such as the paper of ORCA and NACH. The data set is enough to analyze the effectiveness of different SSL methods.
>
> ## R 1.5 Will the additional objectives significantly increase the training time, regarding the speed for convergence?
>
> We just add our designed contrastive learning loss to the original model, which is not computationally expensive, so it does not significantly increase the training time. All the experiments were done based on ORCA on the cifar10 dataset with the same device.
>
> | Running Time     |original  |with CONTROL |
> |----  |----  |----  |
> | For one epoch   |  49.6 seconds   | 52.5 seconds |
>
> We only increased the running time by 5.8 percent from the original algorithm, we gain 4.1 percent performance gain on unseen classes accuracy. It's well worth the time it takes.
>
>
>
>
> ## R 1.6 Abaliton of L_supseen and L_simALL
>
> About the proof provided in Section 4.1, we illustrate the importance of feature-level optimization by whether the optimal classifier learned in the presence of mispairing is consistent with the ideal optimal classifier, Eq. 3 illustrates inconsistency at logits level and Eq. 4 illustrates consistency at feature level.
>
> All the three losses are proposed in the feature-level optimization which have theoretically guaranteed to get a performance gain.
>
> All experiments were done based on NACH on the cifar10 dataset.
> |LSupSeen|	LSimAll|	LSupNN|	Seen|	Unseen	|All|
> |  ----  | ----  |----  |----  |----  |----  |
> |	|	|	|73.5|	50.1|	56.8|
> |√	|	|	|74.0|	50.2|	56.9|
> |	|√	|	|73.6|	50.7|	57.0|
> |	|	|√	|74.0|	51.4|	56.9|
> |√	|√|	√	|74.1|	53.3|	58.0|
>
> As we can see, LSimAll and LSupSeen can each gain performance improvements over existing algorithms.

---

### Meta-Review · Area_Chair_jCrn · 2023-12-12

**Metareview:**

Summary of the paper:

This paper aims to exploit contrastive learning for open-world semi-supervised classification tasks. The key idea involves introducing three loss functions, including nearest-neighbor contrastive learning, supervised contrastive learning, and unsupervised contrastive learning, into existing open-world semi-supervised learning approaches. A notable departure and claimed advantage over prior work is the emphasis on feature-level representation learning, as opposed to logit-level optimization employed in existing BCE loss-based methods, substantiated by supporting theoretical analysis. Experimental evaluations on CIFAR and ImageNet datasets demonstrate that integrating the proposed loss functions with two existing baselines results in enhanced classification accuracy for both seen and unseen classes.

Strengths: The reviewers acknowledge several key strengths in the paper: 1) addressing an important practical task, 2) presenting a straightforward yet effective approach, 3) providing interesting theoretical analysis, and 4) conducting a comprehensive set of experiments with compelling results and informative ablations.

Weaknesses and missing in the submission: While the rebuttal has effectively addressed some of the reviewers' concerns, there remains a consensus among reviewers regarding the weaknesses and limitations of the paper, as outlined below:

1) The primary weakness identified is the lack of technical novelty. The proposed method appears to be a combination of existing contrastive learning loss functions—namely, nearest-neighbor contrastive learning, supervised contrastive learning, and unsupervised contrastive learning. The perceived benefits of these losses do not seem specifically tailored to the task of open-world semi-supervised learning. Reviewers noted that beyond this combination, the paper lacks a distinct technical contribution, and this aspect is not well emphasized in the current submission. Particularly highlighted by the reviewers is the authors' clarification that the technical contribution lies in the natural emergence of the adopted contrastive learning loss functions from the analysis of previous approaches—an aspect that should have been explicitly highlighted and clarified in the submission.

2) The benefits of the proposed strategy appear inconsistent. In the rebuttal, the authors provided an additional result that combines the proposed strategy with the state-of-the-art method OpenCon. However, the outcome reveals a significant drop, suggesting that the proposed strategy adversely affects the performance of OpenCon and is thus not universally applicable. Furthermore, there is a noticeable absence of an in-depth discussion on the reasons behind this observed impact.

3) The sensitivity to hyperparameter settings is evident from the newly provided results in the rebuttal, indicating that the proposed method's performance is influenced by the choice of hyperparameters.

4) Reviewers have expressed concerns about the paper's presentation, noting a lack of emphasis on overall research insights, omission of crucial details, imprecise statements, and sentences containing grammar issues. While the revision has partially improved clarity, substantial rewriting is still necessary to address these issues.

**Justification For Why Not Higher Score:**

Despite the authors’ efforts to enhance the submission and provide clarifications during the discussion stage, the reviewers remained unconvinced and maintained their initial assessment. By the end of the discussion stage, all four reviewers kept their ratings as marginally below the acceptance threshold.

In light of the reviewers’ comments and recommendations, the area chairs did not identify sufficient reason to overturn the consensus reached by the reviewers. The area chairs acknowledge the paper's potential and its implications for practical applications. By addressing the remaining concerns, such as technical novelty, providing an in-depth discussion on the results, and undergoing substantial revision, the paper could significantly improve in the next cycle.

**Justification For Why Not Lower Score:**

N/A

---

### Decision · Program_Chairs · 2024-01-16

Reject